# URBANPLANBENCH: A COMPREHENSIVE URBAN PLANNING BENCHMARK FOR EVALUATING LARGE LANGUAGE MODELS

## ABSTRACT

The advent of Large Language Models (LLMs) holds promise for revolutionizing various fields traditionally dominated by human expertise. Urban planning, a professional discipline that fundamentally shapes our daily surroundings, is one such field, heavily relying on multifaceted domain knowledge and experience of human experts. However, the extent to which LLMs can assist human practitioners in urban planning remains largely unexplored. In this paper, we introduce a comprehensive benchmark, UrbanPlanBench, tailored to evaluate the efficacy of LLMs in urban planning, which encompasses fundamental principles, professional knowledge, and management and regulations, aligning closely with the qualifications expected of human planners. Through extensive evaluation, we reveal a significant imbalance in the acquisition of planning knowledge among LLMs, with even the most proficient models falling short of meeting professional standards. For instance, we observe that 70% of LLMs achieve subpar performance in understanding planning regulations compared to other aspects. Besides the benchmark, we present the largest-ever supervised fine-tuning (SFT) dataset, UrbanPlanText, comprising over 30,000 instruction pairs sourced from urban planning exams and textbooks. Our findings demonstrate that fine-tuned models exhibit enhanced performance in memorization tests and comprehension of urban planning knowledge, while there exists significant room for improvement, particularly in tasks requiring domain-specific terminology and reasoning. By making our benchmark, dataset, and associated evaluation and fine-tuning toolsets publicly available at https://anonymous.4open.science/r/PlanBench, we aim to catalyze the integration of LLMs into practical urban computing, fostering a symbiotic relationship between human expertise and machine intelligence.

## 1 INTRODUCTION

Recent breakthroughs in Large Language Models (LLMs) (Touvron et al., 2023; Zeng et al., 2023) have showcased remarkable capabilities in generating text, reasoning, and knowledge QA, unlocking a plethora of applications ranging from chatbots (OpenAI, 2022) to programming copilots (Chen et al., 2021). Besides general-purpose evaluation, assessing their capabilities in specialized domains is crucial for understanding the real-world impact of LLMs (Chen and Deng, 2023; Koncel-Kedziorski et al., 2023; Fei et al., 2023; Liu et al., 2024). In this paper, we focus on one critical field, urban planning, which stands as a cornerstone in shaping modern city life, yielding profound influence on over 4 billion urban residents worldwide. It is a complex endeavor that intertwines various disciplines, demanding a deep understanding of domain knowledge. Despite the advent of technological advancements, the field continues to heavily rely on the expertise and experience of human planners. For instance, human planners devote substantial time to tasks such as planning text management, review, and assessment (Zhu et al., 2024). Moreover, the limitations inherent in human experience often lead to errors and inefficiencies in planning outcomes (Zheng et al., 2023a).

Notably, the integration of LLMs in urban planning contexts has emerged as a promising avenue, leveraging their pre-trained world knowledge to tackle complex computational tasks (Zhou et al., 2024; Fu et al., 2024; Xu et al., 2023; Zhu et al., 2024; Li et al., 2024). However, the inherent challenges of hallucination and vagueness present significant hurdles, particularly when addressing

specialized problems within urban planning (Zhang et al., 2023). While various benchmarks such as SuperGLUE Wang et al. (2019), BIG-BENCH (Srivastava et al., 2022) and C-Eval (Huang et al., 2023) have been proposed to evaluate LLM effectiveness in understanding and solving intricate tasks, the absence of dedicated benchmarks for urban planning restricts our ability to quantify the extent to which LLMs acquire specialized knowledge and their potential to enhance the productivity of human planners. This gap underscores the need for a comprehensive benchmark tailored to evaluate LLM performance in urban planning domains.

As a human-centered field, matching the performance of human planners marks a milestone for LLMs and signifies their mastery of urban planning capabilities. In alignment with the rigorous standards set by the certified urban planner qualification examination in China, we introduce UrbanPlanBench, a comprehensive benchmark designed to evaluate LLMs across various perspectives of urban planning, including fundamental principles, professional knowledge, and management and regulations. The benchmark mirrors the latest available examination standards as of 2022, enabling a comparative analysis between LLM and human planners, shedding lights on whether current general-purpose LLMs attain a human-level understanding of urban planning. Leveraging this benchmark, we scrutinize recent open-source LLMs, including LLaMA1/2/3/3.1 (Touvron et al., 2023; Meta, 2024), Gemma1/2 Team et al. (2024a;b), ChatGLM3/4 GLM et al. (2024), Baichuan2 (Baichuan, 2023), Qwen1.5/2 (Bai et al., 2023; Yang et al., 2024), and Yi (Young et al., 2024), as well as commercial LLMs like ChatGPT 3.5/4o, to assess their acquisition of planning skills. Additionally, we also evaluate the effect of prompting techniques for LLMs including chain of thought (COT) Wei et al. (2022b) and retrieval augmented generation (RAG) Lewis et al. (2020); Gao et al. (2023).

Supervised fine-tuning (SFT) stands as a prevalent method to build domain-specific LLMs. However, to our knowledge, there are currently no off-the-shelf resources for fine-tuning LLMs specifically for urban planning. This gap arises from the significant disparity between the distribution of urban planning knowledge in descriptive texts and the required form of SFT data, which necessitates sample pairs comprising instructions and responses. To bridge this gap, we further introduce UrbanPlanText, the largest-ever dataset tailored for SFT of LLMs in urban planning. Comprising over 30,000 instruction pairs derived from textbooks and past exams, UrbanPlanText serves as a comprehensive collection of specialized urban planning content.

We conduct extensive experiments to assess current advanced LLMs on UrbanPlanBench. While LLMs demonstrate significantly better performance than random guessing, there remains large room for improvement, indicating a limited mastery of urban planning skills. Notably, most of the current LLMs can not surpass the certification bar of the urban planner qualification examination, which roughly represents the top 10% proficiency level of human planners. Additionally, our analysis highlights an imbalance in LLM performance across three key urban planning perspectives: they exhibit greater proficiency in understanding planning principles and knowledge but tend to falter in memorizing regulations, leading to factual errors. Moreover, we find that finetuning LLMs with UrbanPlanText can effectively enhance their ability to answer urban planning-related questions. By introducing both UrbanPlanBench and UrbanPlanText, we aim to facilitate the seamless integration of LLMs into practical urban planning workflows, thereby removing barriers for human practitioners and empowering them to harness the potential of advanced AI tools in their endeavors.

## 2 BENCHMARKING LLMs ON URBAN PLANNING

### 2.1 CONSTRUCTION OF URBANPLANBENCH

The motivation for UrbanPlanBench stems from the need to quantitatively assess the extent to which LLMs acquire expertise in urban planning. Specifically, we aim to answer the fundamental question of whether LLMs can match the proficiency of human planners, given that urban planning is inherently a human-centered field. To achieve this goal, we have constructed UrbanPlanBench based on the latest available real-world urban planning qualification exam in China, which serves as the standard for certifying registered urban planners. This benchmark evaluates LLMs from the following three critical perspectives (subjects) of urban planning:

- **S1: Fundamental principles.** This subject delves into topics concerning cities and urban development, basic know-how of urban planning, urban land use and spatial layout, as well as practical

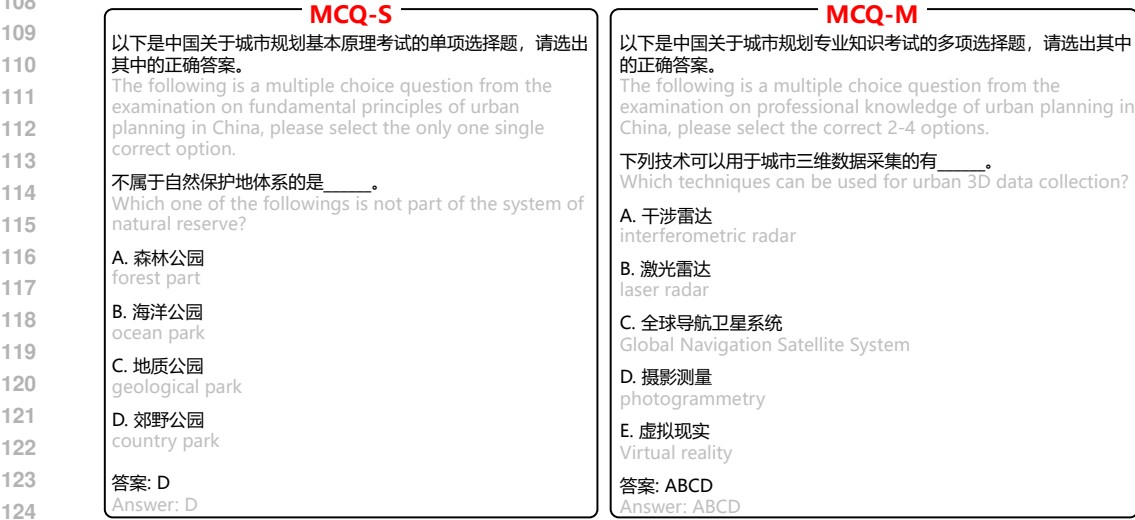

Figure 1: Example questions of UrbanPlanBench. MCQ-S has four options where only one option is correct. MCQ-M is more challenging, featuring two to four correct options from a total of five options.

implementation of urban planning. It reflects LLMs' grasp of the foundational theories underlying urban development and the discipline of urban planning.

- **S2: Professional knowledge.** This subject covers knowledge from eight professional fields that are closely related to urban planning, which include architecture, urban transportation, municipal public facilities, information technology application in urban planning, urban economics, urban geography, urban sociology, and urban ecology and environment. It measures LLMs' proficiency, familiarity, and comprehension across various disciplines relevant to urban planning.

- **S3: Management and regulations.** Management covers urban planning formulation and approval management, implementation management, supervision and inspection, and professional ethics. Regulations include foundational knowledge in administrative and urban planning laws, complementary regulations, technical standards and specifications, and relevant laws and policies.

By incorporating the above diverse perspectives, UrbanPlanBench forms a challenging testbed that comprehensively evaluates the mastery of urban planning skills for LLMs, shedding light on their capabilities in this complex domain.

In constructing UrbanPlanBench, we adopted the widely used multiple-choice question (MCQ) format (Hendrycks et al., 2020), due to its efficacy in assessing LLMs' understanding and reasoning capabilities, with rigorously defined accuracy. For each of the three aforementioned perspectives, we crafted 100 MCQs. In each category, the initial 80 MCQs feature four choices, with only one correct answer. To further challenge the LLMs, the remaining 20 MCQs in each perspective include five choices, with two to four correct options. All questions in UrbanPlanBench were curated from PDF or Microsoft Word documents, and meticulously transformed into a structured CSV format through careful parsing and annotation by the authors. These questions were presented to the LLMs through prompts, as demonstrated in Figure 1.

In the context of the urban planner qualification exams, achieving a score of 60 out of 100 MCQs correctly answered across all subjects stands as a crucial criterion for certification. These exams pose a significant challenge even for human participants, with only 10% passing annually. Therefore, if LLMs can consistently answer 60 MCQs correctly across the three subjects, it suggests they have attained a level of urban planning expertise comparable to that of registered human planners, signifying the top 10% of human-level proficiency. By adhering to the rigorous standards set by real-world examinations, our constructed benchmark aims to offer tangible insights into the capabilities of LLMs that can be directly compared with those of human planners. Subsequently, we evaluate

Table 1: Accuracy (%) of LLMs on three subjects of UrbanPlanBench. S and M indicate MCQs with one single correct answer and multiple correct answers, respectively. Full represents the overall accuracy of both types of MCQs.

| Model | S1 | | | S2 | | | S3 | | |
|---|---|---|---|---|---|---|---|---|---|
| | S | M | Full | S | M | Full | S | M | Full |
| Random | 25.0 | 4.0 | 20.8 | 25.0 | 4.0 | 20.8 | 25.0 | 4.0 | 20.8 |
| LLaMA-7B | 28.8 | 15.0 | 26.0 | 27.5 | 5.0 | 23.0 | 21.3 | 0.0 | 17.0 |
| LLaMA-13B | 25.0 | 15.0 | 23.0 | 26.3 | 5.0 | 23.0 | 23.8 | 0.0 | 19.0 |
| LLaMA-30B | 26.3 | 15.0 | 24.0 | 25.0 | 5.0 | 22.0 | 32.5 | 0.0 | 26.0 |
| LLaMA2-13B | 27.5 | 15.0 | 25.0 | 28.8 | 5.0 | 24.0 | 20.0 | 0.0 | 16.0 |
| LLaMA3-8B-base | 42.5 | 10.0 | 36.0 | 53.7 | 20.0 | 47.0 | 37.5 | 0.0 | 30.0 |
| LLaMA3-8B-chat | 46.3 | 15.0 | 40.0 | 58.8 | 25.0 | 52.0 | 38.8 | 0.0 | 31.0 |
| LLaMA3.1-8B | 56.3 | 10.0 | 47.0 | 46.3 | 10.0 | 39.0 | 43.8 | 0.0 | 35.0 |
| LLaMA3.1-70B | 42.5 | 10.0 | 36.0 | 53.8 | 20.0 | 47.0 | 37.5 | 0.0 | 30.0 |
| LLaMA3.1-405B | 48.8 | 5.0 | 40.0 | 41.3 | 5.0 | 34.0 | 47.5 | 10.0 | 40.0 |
| Gemma-7B | 26.3 | 5.0 | 22.0 | 22.5 | 0.0 | 18.0 | 27.5 | 0.0 | 22.0 |
| Gemma2-9B | 23.8 | 5.0 | 20.0 | 21.3 | 0.0 | 17.0 | 27.5 | 5.0 | 23.0 |
| GPT-3.5-turbo | 51.3 | 0.0 | 41.0 | 53.8 | 15.0 | 46.0 | 32.5 | 15.0 | 29.0 |
| GPT-4o-mini | 35.0 | 5.0 | 29.0 | 40.0 | 10.0 | 34.0 | 35.0 | 10.0 | 30.0 |
| ChatGLM3-6B-base | 47.5 | 5.0 | 39.0 | **60.0** | 25.0 | 53.0 | 50.0 | 5.0 | 41.0 |
| ChatGLM3-6B-chat | 38.8 | 5.0 | 32.0 | 51.3 | 30.0 | 47.0 | 41.3 | 5.0 | 34.0 |
| ChatGLM4-9B | 56.3 | 10.0 | 47.0 | **73.8** | 5.0 | **60.0** | **61.3** | 10.0 | 51.0 |
| Baichuan2-7B-base | 50.0 | 5.0 | 41.0 | 47.5 | 0.0 | 38.0 | 38.8 | 15.0 | 34.0 |
| Baichuan2-7B-chat | 36.3 | 5.0 | 30.0 | 51.3 | 25.0 | 46.0 | 40.0 | 0.0 | 32.0 |
| Qwen1.5-7B-base | 53.8 | 15.0 | 46.0 | **60.0** | 10.0 | 50.0 | 53.8 | 10.0 | 45.0 |
| Qwen1.5-7B-chat | 47.5 | 15.0 | 41.0 | 63.8 | 15.0 | 54.0 | 48.8 | 5.0 | 40.0 |
| Qwen1.5-110B | **60.0** | 15.0 | 51.0 | **82.5** | 35.0 | **73.0** | 63.8 | 45.0 | **60.0** |
| Qwen2-7B | **66.3** | 15.0 | 56.0 | **70.0** | 25.0 | **61.0** | 65.0 | 10.0 | 54.0 |
| Qwen2-70B | **70.0** | 30.0 | **62.0** | **77.5** | 45.0 | **71.0** | 68.8 | 45.0 | **64.0** |
| Yi-6B-base | **61.3** | 15.0 | 52.0 | **65.0** | 5.0 | 53.0 | **60.0** | 10.0 | 50.0 |
| Yi-6B-chat | **62.5** | 0.0 | 50.0 | **70.0** | 30.0 | **62.0** | 56.3 | 5.0 | 46.0 |
| Cert. Bar (top 10% human) | - | - | 60.0 | - | - | 60.0 | - | - | 60.0 |

a diverse array of advanced LLMs on this benchmark to comprehensively scrutinize their urban planning abilities.

## 2.2 EVALUATION RESULTS

In our experimental evaluation, we prompted multiple advanced LLMs to respond to all questions presented in the introduced UrbanPlanBench. For each question, we selected the option with the highest output probability by each LLM as its final response (Hendrycks et al., 2020), and then calculated the average accuracy within different subjects. Table 1 illustrates the benchmarking results of LLMs, detailing the accuracy for both the 80 MCQ-S (single correct option) questions and the 20 MCQ-M (multiple correct options) questions separately, along with the accuracy for the entire set of 100 questions within each subject. We have the following empirical findings:

- *Overall planning capabilities.* (1) We observe that current advanced LLMs demonstrate a substantial level of proficiency in urban planning expertise. Across all three subjects, the accuracy rates of all LLMs notably surpass random guess predictions, indicating the effectiveness of large-scale pretraining and supervised fine-tuning in equipping these models with urban planning memorization and reasoning abilities. Specifically, the highest-performing LLM achieves approximately 2.98, 3.51, and 3.08 times higher accuracy than random guess predictions in fundamental principles, professional knowledge, and management and regulations, respectively. Moreover, we observe that 9 LLMs achieve at least 50.0% accuracy in at least one subject, underscoring their mastery of urban planning expertise. (2) However, despite these promising results, LLMs still lag significantly behind professional human planners in terms of performance. All 25 evaluated LLMs, except

for Qwen2-70B, fail to exceed the certification bar for professional human planners, *i.e.* 60.0% accuracy in all three subjects. Specifically, out of 75 cases comprising 25 different LLMs and 3 subjects, only 8 times does an LLM exceed the 60% accuracy certification bar which roughly aligns with top 10% human proficiency levels. This indicates that most of the LLMs evaluated in this study are not capable of passing the urban planning qualification exam, highlighting their insufficient urban planning capabilities compared to certified human planners. (3) Furthermore, we find that LLMs perform notably worse on MCQ-M questions compared to MCQ-S questions. This discrepancy is understandable, given the increased complexity of MCQ-M questions, which feature a set of 25 potential answers, much larger than MCQ-S questions that only have 4 potential answers. Specifically, we observe zero accuracy in 15 out of 75 cases, indicating a performance level even below random guess predictions. These findings suggest that, while most existing benchmarks for LLMs primarily focus on MCQ-S questions, it may be necessary to include more challenging MCQ-M tests to comprehensively evaluate the capabilities of LLMs in specialized domains such as urban planning.

- *Subject imbalance.* The results in Table 1 reveal an obvious imbalance in the performance of LLMs across the three distinct subjects evaluated in UrbanPlanBench. Specifically, we find that the average accuracy of the 25 LLMs on the three subjects is 38.24%, 44.16%, and 34.52%, respectively. Particularly, LLMs demonstrate significantly better performance on S2 (professional knowledge) compared to the other two subjects, S1 (fundamental principles) and S3 (management and regulations). Moreover, 68% LLMs achieve accuracy lower than 45.0% in both S1 and S3, and only 16% models achieves over 50.0% accuracy in these two subjects. Upon closer examination of the definitions of the three subjects, we observe that S2 covers a broader range of general and diverse topics, potentially overlapping with the pretraining and SFT data of these LLMs. In contrast, S1 and S3 focus more on domain-specific contents, emphasizing specialized urban planning concepts that may be insufficiently represented in the training data. These findings underscore the need to develop a specialized SFT dataset tailored specifically to urban planning to enhance the performance of LLMs in this critical domain.

- *Language bias.* UrbanPlanBench is a Chinese benchmark sourced from questions of urban planning exams in China, thus most of the evaluated LLMs are also Chinese LLMs which are pre-trained and finetuned with large-scale Chinese textual data. Still, we include three English-primary LLM series for comparison, namely LLaMA, Gemma, and GPT. The results highlight a notable difference between the performance of Chinese LLMs and two English-primary LLMs, particularly evident in S3 (management and regulations). The average accuracy of the three English-primary LLM series in S3 is 26.77%, representing a significant 41.7% relative gap compared to the other eight Chinese LLMs, which exhibit an average accuracy of 45.92%. However, the disparity in performance between the English-primary LLMs and other Chinese LLMs is less pronounced in S1 and S2. For example, the LLaMA3.1-8B and LLaMA3-8B-chat model surpass 7 and 4 Chinese LLMs in terms of accuracy in S1 and S2, respectively. These results suggest a potential difference in the adaptability of English-primary LLMs compared to their Chinese counterparts in comprehending and interpreting the specific regulations and management aspects inherent in urban planning contexts, emphasizing the importance of considering language-specific nuances in LLM performance evaluation and application.

- *Scaling effect.* Researchers have consistently observed scaling laws of neural language models, particularly LLMs, where scaling up models can lead to substantial performance improvements (Kaplan et al., 2020) and even emergent abilities (Wei et al., 2022a). From the results we can observe that larger models generally achieve higher accuracy. For instance, LLaMA3.1-405B improved the performance by 53.8%, 47.8%, and 135.3% on S1, S2, and S3, respectively, in comparison to LLaMA-7B. To further investigate this phenomenon, we evaluated Qwen1.5 models of varying parameter scales, ranging from 0.8B to 32B parameters, on UrbanPlanBench. We calculated accuracy across three subjects, with MCQ-S and MCQ-M questions examined separately. The results, depicted in Figure 2, showcase a notable scaling effect, particularly evident in MCQ-S questions, aligning with previous literature. Across all three subjects, we observed approximately a 100% increase in accuracy for MCQ-S questions by scaling up models, with the largest improvement of 108.0% seen in S1, followed by 96.3% and 89.7% improvements in S3 and S2, respectively. Remarkably, the Qwen1.5-14B and Qwen1.5-32B models achieved over 60% accuracy in two of the three subjects, signaling their potential to rival professional human planners. These findings underscore the validity of scaling laws in LLMs where larger models demonstrate enhanced understanding and reasoning capabilities, as evidenced in our specialized benchmark. However, we also

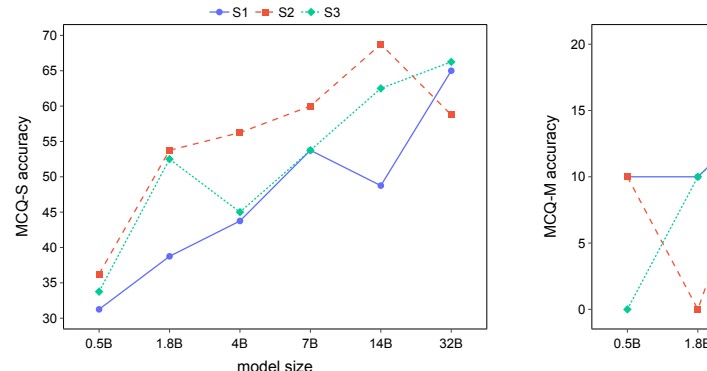

Figure 2: Performance of different model sizes. The LLM model Qwen1.5 is adopted. MCS-S and MCQ-M indicate MCQs with one single correct answer and multiple correct answers, respectively. **Left.** Accuracy on MCQ-S questions. **Right.** Accuracy on MCQ-M questions.

Table 2: Accuracy (%) of LLMs on three subjects of UrbanPlanBench using RAG and CoT techniques. S and M indicate MCQs with one single correct answer and multiple correct answers, respectively. Full represents the overall accuracy of both types of MCQs.

| Model | S1 | | | S2 | | | S3 | | |
|---|---|---|---|---|---|---|---|---|---|
| | S | M | Full | S | M | Full | S | M | Full |
| GPT-4o-mini | 40.0 | 5.0 | 33.0 | 45.0 | 5.0 | 37.0 | 42.5 | 15.0 | 37.0 |
| RAG_exam1 | 52.5 | 5.0 | 43.0 | 68.8 | 45.0 | 64.0 | 53.8 | 5.0 | 48.0 |
| RAG_exam2 | 58.8 | 5.0 | 48.0 | 70.0 | 30.0 | 62.0 | 56.3 | 20.0 | 49.0 |
| COT_exam1 | 60.0 | 15.0 | 51.0 | 70.0 | 35.0 | 63.0 | 58.8 | 25.0 | 53.0 |
| COT_exam2 | 53.8 | 10.0 | 45.0 | 72.5 | 35.0 | 65.0 | 51.3 | 35.0 | 48.0 |
| Cert. Bar (top 10% human) | - | - | 60.0 | - | - | 60.0 | - | - | 60.0 |

observed that accuracy on MCQ-M questions remained low despite increasing model sizes. Given the complexity of MCQ-M tests, merely scaling up LLMs may prove insufficient, necessitating advanced techniques such as retrieval augmented generation (RAG) (Lewis et al., 2020) to bolster the urban planning expertise of LLMs for addressing intricate MCQ-M questions.

- *Longitudinal studies.* To track the evolution of LLM performance over time, we compare different generations of LLMs. We can observe that later generations of LLMs indeed achieve substantially better performance than their corrsponding earlier version in most cases. For example, LLaMA3.1-8B improves the accuracy on S1 by 105% against LLaMA-13B, Qwen2-7B improves the accuracy on S2 by 17.3% against Qwen1.5-7B, and ChatGLM4-9B improves the accuracy on S3 by 13.3% against ChatGLM3-6B. The longitudinal studies confirm that the progress made in data quality, model structure, and training algorithm effectively enhances the ability of LLMs to understand and deal with complex urban planning problems. Nevertheless, the enhanced model capabilities alone are still not effective in dealing with MCQ-M questions, indicating the necessity of incorporating advanced techniques such as COT.

- *Human-Level Performance.* Surprisingly, we find that Qwen2-70B achieved accuracy of 62.0%, 71.0%, and 64.0% on the three subjects, making it the first and only LLM to surpass the 60% certification threshold of professional human planners. The inspiring results highlight the huge potential of LLMs to assist human planners in practical urban planning tasks.

### 2.3 PROMPTING TECHNIQUES

Prompting techniques can significantly enhance LLMs in answering complicated questions and improving their reasoning capabilities. Here in UrbanPlanBench, we use GPT-4o-mini as the base model to validate the effectiveness of RAG Lewis et al. (2020) and CoT Wei et al. (2022b) for knowledge augmentation of LLMs, as well as to evaluate their enhancement of LLMs' competence

Table 3: Statistics of different sources for UrbanPlanText.

| Category | Name | #Words | #Samples |
|---|---|---|---|
| Past Exams | MCQ | 619,810 | 2,397 |
| | dialog | 350,080 | 4,139 |
| Textbooks | *Principles of urban planning* | 470,621 | 5,091 |
| | *Knowledge of urban planning* | 457,236 | 9,307 |
| | *Urban planning management and regulations* | 313,246 | 4,347 |
| | *Urban planning practice* | 120,155 | 3,589 |
| | *Detailed regulatory plan* | 174,626 | 246 |
| | *History of urban construction in China* | 156,923 | 608 |
| | *Additional contents of urban planning exams* | 60,371 | 1,610 |
| | Sum | 2,723,068 | 31,334 |

in the field of urban planning. Table 2 illustrates the results of different prompting techniques. Specifically, we utilize Self-RAG Asai et al. (2023) to retrieve matches in RAG experiments, where RAG_exam1 denotes that the relevant textbooks and previous years' questions are directly used as the content of the knowledge base, and RAG_exam2 denotes that these contents are first processed into high-quality QA. CoT_exam1 and CoT_exam2 denote the few-shot-CoT and zero-shot-CoT. From these experimental results, we have the following observations:

- *RAG prompting.* The introduction of the RAG technique significantly improves the performance on each subject. Specifically, RAG_exam1 and RAG_exam2 improve the accuracy on S1 by 30.3% and 45.5%, respectively, compared to the GPT-4o-mini base model. In S2, RAG_exam1 and RAG_exam2 achieve an overall average accuracy of 64% and 62%, respectively, and both MCQ-S and MCQ-M are improved by more than 52.9%, reaching the level of professional urban planners. In MCQ-M of S3, the GPT-4o-mini base model performs better than RAG_exam1, but the accurate information retrieval still ensures a high accuracy rate in MCQ-S for RAG models with about 32.5% improvements against the base model.

- *CoT prompting.* The introduction of CoT technology leads to stronger reasoning ability of LLMs and significantly improves the performance of each subject. Specifically, CoT_exam1 and CoT_exam2 improved the accuracy by 54.5% and 36.4% in S1, 70.3% and 75.7% in S2, and 43.2% and 29.7% in S3, respectively. Notably, COT substantially improves the performance of LLMs in answering MCQ-M questions. For example, COT_exam2 increases the accuracy on MCQ-M by 100%, 600%, and 133% in the three subjects. As there exist more than one correct options in MCQ-M questions which are much more complicated than MCQ-S ones, the above results confirm the effectiveness of COT in boosting the reasoning ability of LLMs.

  In practical urban planning scenarios, LLMs can be smoothly integrated into planners' workflow using appropriate prompts, and we provide two example cases in Appendix A.2.

## 3 FINE-TUNING LLMS WITH URBANPLANTEXT

The inherent knowledge of LLMs proves insufficient when confronted with specialized urban planning queries, highlighting a deficiency in domain-specific understanding. SFT emerges as a widely adopted technique for tailoring LLMs towards specific domains by infusing them with related knowledge and data. Notably, SFT datasets for LLMs typically consist of sample pairs comprising instructions and corresponding responses. However, existing urban planning knowledge is scattered across unannotated textual resources, presenting a challenge in sourcing relevant data for SFT. Towards this end, we initially gathered materials from seven urban planning textbooks, along with archives of urban planning exams spanning the past eight years. Subsequently, we derive instruction pairs from these materials, leading to the largest-ever SFT dataset tailored for urban planning.

### 3.1 DATASET CONSTRUCTION

**Data Sources.** Our urban planning textual data collection primarily focuses on two key sources: urban planning textbooks and past urban planning exams, with the overview of the dataset's statistics shown

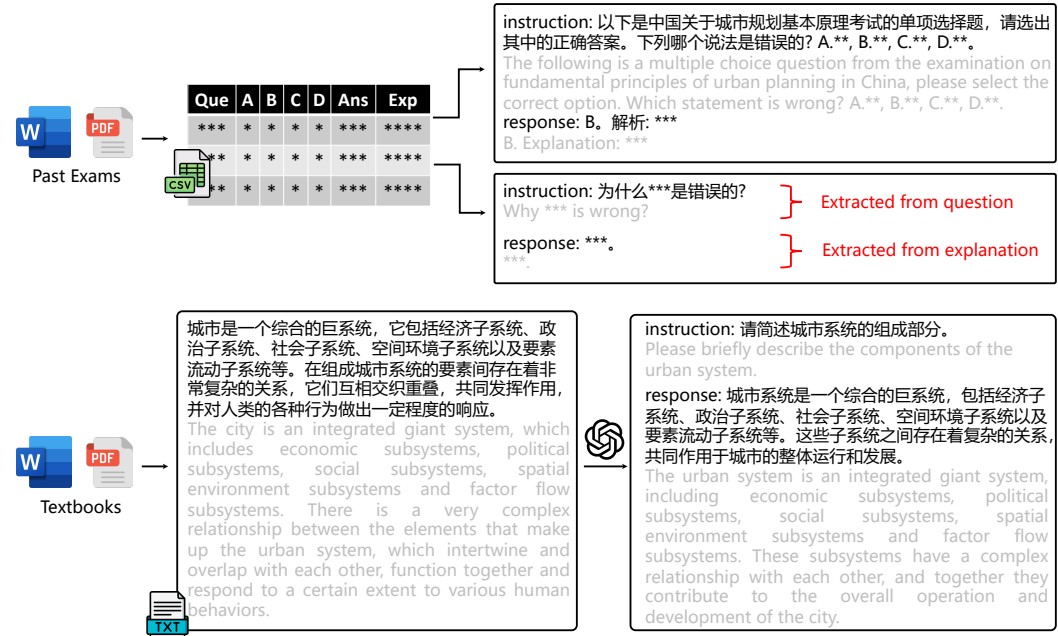

Figure 3: Data collection and process of UrbanPlanText. Past exam questions are first annotated by the authors into structured CSV files and then transformed into MCQ-type instruction pairs and dialog-style instruction pairs. Textbooks are first parsed into textual files, from which instruction pairs are generated automatically by prompting OpenAI's ChatGPT model.

in Table 3. Urban planning textbooks encompass a wide spectrum of general knowledge about urban planning, thus leveraging data from textbooks enables LLMs to establish a foundational understanding of the specific domain, mirroring the approach taken by human planners who frequently refer to textbooks as their primary learning and training resources. Conversely, questions found in urban planning exams adhere to specific formats and emphasize particular areas. Therefore, fine-tuning LLMs with data extracted from real exams serves to further refine their abilities, enhancing their accuracy in addressing domain-specific exam questions. Particularly, as the introduced benchmark UrbanPlanBench is sourced from the latest publicly available urban planning exam in China held in 2022, we utilize exam questions predating 2022 for the collection of SFT data, spanning eight years.

**Data Processing.** The collected original materials encompass a variety of formats, predominantly stored as PDF or Microsoft Word documents. To facilitate further processing, these materials are first parsed into plain text format. In the case of urban planning exams from previous years, an additional step is taken to transform these MCQs into a structured CSV format before they are processed into instruction pairs. These instruction pairs are designed to include the system prompt, the question itself, and the provided options, while the response comprises the correct answer accompanied by explanations. Meanwhile, dialog-style instruction pairs are derived from MCQs to further enrich the training materials, as depicted in Figure 3. However, for urban planning textbooks, which primarily contain descriptive text without readily available instruction pairs, more data process is needed. Here, we employ OpenAI's ChatGPT to automatically generate instruction pairs from the descriptive text by prompting, as demonstrated in Figure 3. This process ensures the conversion of all collected materials into a standardized format suitable for subsequent SFT. Details regarding the quality of the generated data is provided in Appendix A.1.

## 3.2 SFT RESULTS

We leveraged our constructed UrbanPlanText dataset to fine-tune LLMs using LLaMA-Factory (Zheng et al., 2024). Employing lora (Hu et al., 2021) to accelerate the SFT process, we fine-tuned all models for three epochs on one single Nvidia A100 GPU, which takes about 4 hours. Subsequently, we evaluated the fine-tuned LLMs again on UrbanPlanBench, with the results detailed in Table 4.

Table 4: Accuracy (%) of LLMs on three subjects of UrbanPlanBench after SFT on UrbanPlanText. S and M indicate MCQs with one single correct answer and multiple correct answers, respectively. Full represents the overall accuracy of both types of MCQs. **Bold numbers** indicate that the performance improves against the corresponding pre-SFT model.

| Model | S1 | | | S2 | | | S3 | | |
|---|---|---|---|---|---|---|---|---|---|
| | S | M | Full | S | M | Full | S | M | Full |
| LLaMA3-8B-base | 40.0 | 5.0 | 33.0 | 57.5 | 25.0 | **51.0** | 41.3 | 0.0 | **33.0** |
| LLaMA3-8B-chat | 42.5 | 10.0 | 36.0 | 51.3 | 10.0 | 43.0 | 33.8 | 5.0 | 28.0 |
| ChatGLM3-6B-base | 50.0 | 5.0 | **41.0** | 58.8 | 35.0 | **54.0** | 55.0 | 5.0 | **45.0** |
| ChatGLM3-6B-chat | 35.0 | 5.0 | 29.0 | 52.5 | 25.0 | 47.0 | 42.5 | 5.0 | **35.0** |
| Baichuan2-7B-base | 46.3 | 5.0 | 38.0 | 56.3 | 5.0 | **46.0** | 43.8 | 15.0 | **38.0** |
| Baichuan2-7B-chat | 43.8 | 5.0 | **36.0** | 46.3 | 20.0 | 41.0 | 31.3 | 0.0 | 25.0 |
| Qwen1.5-7B-base | 50.0 | 10.0 | 42.0 | 63.8 | 5.0 | **52.0** | 53.8 | 10.0 | 45.0 |
| Qwen1.5-7B-chat | 48.8 | 15.0 | **42.0** | 63.8 | 5.0 | 52.0 | 50.0 | 10.0 | **42.0** |
| Yi-6B-base | 58.8 | 15.0 | 50.0 | 68.8 | 0.0 | **55.0** | 61.3 | 0.0 | 49.0 |
| Yi-6B-chat | 62.5 | 0.0 | 50.0 | 66.3 | 35.0 | 60.0 | 63.8 | 0.0 | **51.0** |

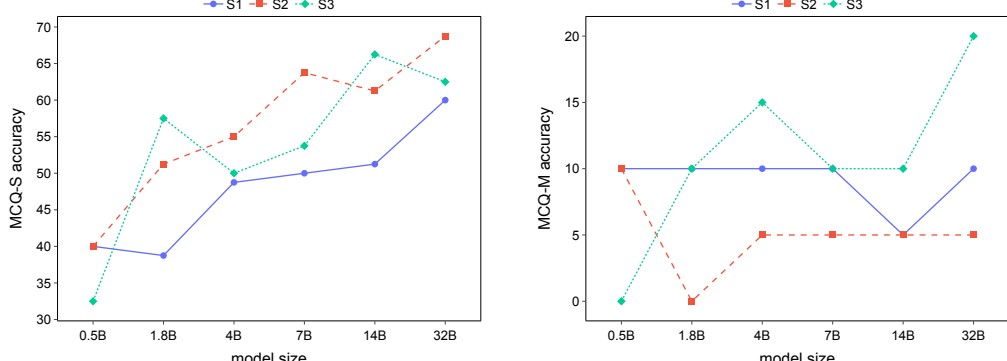

Figure 4: Performance of different model sizes after SFT on UrbanPlanText. The LLM model Qwen1.5 is adopted. MCS-S and MCQ-M indicate MCQs with one single correct answer and multiple correct answers, respectively. **Left.** Accuracy on MCQ-S questions. **Right.** Accuracy on MCQ-M questions.

Notably, we observed a significant enhancement in performance, particularly in S3 (management and regulations). Specifically, 60% of LLMs exhibited improved accuracy on full questions of S3, and 70% demonstrated enhanced accuracy on the MCQ-S questions of S3, with the average accuracy of LLMs improved by 2.1% compared to their pre-SFT counterparts. Given that S3 was previously the weakest subject for LLMs according to Table 1, these findings underscore the effectiveness of enhancing the domain-specific capabilities of LLMs through SFT. Additionally, for the previously strongest subject, S2 (professional knowledge), LLMs maintained competitive performance, with an average accuracy of 50.1%, similar to the pre-SFT average accuracy of 50.2%.

Additionally, we conducted SFT experiments on LLMs of varying sizes using UrbanPlanText and subsequently evaluated their performance on UrbanPlanBench. Aligning with previous benchmarking experiments, we still employed the Qwen1.5 model across a spectrum of parameter sizes ranging from 0.8B to 32B. We illustrated their post-SFT performance in Figure 4. Similar to previous observations, we can observe a clear scaling effect on the accuracy of MCQ-S questions, with larger models demonstrating substantially improved performance compared to their smaller counterparts. Particularly, we noted that SFT yielded more substantial benefits for smaller models. For instance, the average accuracy on MCQ-S questions across all three subjects increased by 11.1%, 1.7%, and 6.0% for the smallest three models (0.5B, 1.8B, and 4B) compared to previous results in Figure 2, respectively, while the improvement was only 0.7% for the largest 32B model. These findings hold significant practical implications, particularly as smaller models are more accessible to a broader user base at a considerably lower cost.

## 4    RELATED WORK

**AI for urban planning.** AI applications in urban planning offer promising solutions to the challenges posed by rapid urbanization, aiming to alleviate the burden on human planners. Current research predominantly focuses on urban design, generating layouts of various urban functionalities such as land use (Zheng et al., 2023a), transportation networks (Zheng et al., 2023b; Su et al., 2024), buildings (Qin et al., 2024), and points of interest (POIs) (Wang et al., 2020). These endeavors approach urban design as either a generation problem, utilizing existing urban data and generative models like diffusion models (Qin et al., 2024) and generative adversarial networks (GANs) (Wang et al., 2020), or as an optimization problem tackled through methods such as reinforcement learning (RL) (Zheng et al., 2023a;b; Su et al., 2024) to find more efficient layouts. Despite urban design, urban planners still devote significant time to handling urban planning-related text. The emergence of LLMs has led to the development of specialized models tailored for urban planning tasks (Wang et al., 2024; Zhang et al., 2024; Zhu et al., 2024). For instance, TransGPT (Wang et al., 2024) fine-tunes LLMs with large-scale transportation text to assist in transportation planning, while PlanGPT (Zhu et al., 2024) equips LLMs with external knowledge and web search capabilities for various text-related tasks in urban planning. However, these efforts often rely on case studies to demonstrate the effectiveness of LLMs in urban planning, underscoring the urgent need for a comprehensive benchmark to quantitatively assess the extent to which LLMs masters urban planning knowledge.

**Domain-specific benchmarks for LLMs.** Benchmarks play a pivotal role in shaping the trajectory of AI research, serving as foundational tools that drive progress within the field (Patterson, 2012). LLMs have demonstrated exceptional natural language understanding, reasoning, and memorization abilities, as evidenced by benchmarks such as SuperGLUE (Wang et al., 2019), BIG-Bench (Srivastava et al., 2022), MMLU (Hendrycks et al., 2020), and HELM (Liang et al., 2022), which cover diverse Natural Language Processing (NLP) tasks. While general-purpose NLP benchmarks have provided valuable insights into LLM capabilities, domain-specific benchmarks are indispensable to understand LLMs' specialized expertise (Chen and Deng, 2023; Koncel-Kedziorski et al., 2023; Fei et al., 2023; Liu et al., 2024). Examples include LawBench (Fei et al., 2023), which evaluates LLMs' legal capabilities in memorization, understanding, and application of legal knowledge, and BizBench (Koncel-Kedziorski et al., 2023), which assesses LLMs' ability to reason about financial problems and synthesize code to accomplish Q&A tasks over financial data. Additionally, MathBench (Liu et al., 2024) evaluates LLMs' mathematical proficiency in answering theoretical questions and solving application problems. However, within the realm of urban planning, there is a notable absence of publicly available benchmarks, impeding the effective utilization of LLMs in this critical domain. In response to this gap, this paper proposes the first urban planning benchmark for LLMs, aiming to comprehensively evaluate their capabilities and guide technological advancements in this field.

## 5    CONCLUSION AND FUTURE WORK

This paper introduces UrbanPlanBench and UrbanPlanText, the first urban planning benchmark and the largest-ever SFT dataset tailored for LLMs. These resources, along with open-sourced toolsets, provide comprehensive support for fine-tuning and evaluating LLMs in the critical domain of urban planning. Through a series of experiments involving multiple advanced LLMs, we have showcased their remarkable capabilities in mastering urban planning knowledge. However, there remains substantial untapped potential to fully leverage LLMs to enhance the productivity of human practitioners in this field. We envision that our findings will foster interdisciplinary collaboration between human planners and AI practitioners, paving the way for further exploration and the application of LLMs in influential real-world urban planning scenarios. Moving forward, our future work includes expanding both UrbanPlanBench and UrbanPlanText to incorporate multi-linguistic urban planning materials, thereby enabling broader use cases of the benchmark and dataset. Additionally, we aim to extend UrbanPlanBench into a multi-modal benchmark, integrating both imagery of urban plans and their corresponding descriptive text, further enriching the evaluation capabilities of LLMs in urban planning contexts.

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

## A    APPENDIX

### A.1    SFT DATA QUALITY

In our experiments, we tried different approaches to generate instruction pairs including OpenAI's ChatGPT, Ernie, and the opensourced Bonito framework. Eventually we employed OpenAI's ChatGPT due to its better performance. The prompt template of data generation is as follows:

`#prompt`

#01 You are a Q&A pair dataset processing expert.

#02 Your task is to generate corresponding Q&A pairs based on my questions and the content I give.

#03 The questions generated must be macro and value based, don't generate particularly detailed questions, and not too long.

#04 Answers must be comprehensive, use more of my information and be more informative.

#05 The text is coherent and the content is complete.

#06 Use standardised language, official and rigorous content, no colloquial expressions, no English, pinyin, internet terms, etc.

#07 Beautiful language, rigorous content, no concepts and repetitive content.

#08 Must be generated according to the following sample format:

"instruction": "question",

"output": "answer"

#09 The reference case is as follows:

"instruction": "How can town system planning contribute to the sustainable development of a town system?" ,

"output": "Town system planning to promote the sustainable development of the town system needs to consider comprehensively from the aspects of resource utilisation, environmental protection, social and economic development."

Generate 20 Q&A pairs, ending with the full JSON formatted Q&A pair, discarding contexts that do not respond to completion.

```
(The above contents are translated from Chinese)
```

It is worthwhile to notice that using LLMs to generate training data has become a common practice proven to be effective and widely adopted by related literature Betker et al. (2023); Esser et al. (2024). To verify the data quality, we invited five domain experts with graduated degree in urban planning to judge the generated instruction pairs, comparing different approaches. Specifically, we sample 100 instruction pairs generated by different approaches, and asked the domain experts to score the generated data from the perspective of both correctiveness and informativeness. The results are demonstrated in Table 5.

| Expert | Metric | Model | | |
|---|---|---|---|---|
| | | ChatGPT | Ernie | Bonito |
| 1 | Correctiveness | 9.6 | 9.6 | 7.2 |
| | Informativeness | 9.4 | 8.8 | 5.8 |
| 2 | Correctiveness | 9.2 | 7.4 | 7.6 |
| | Informativeness | 8.4 | 6.8 | 7.6 |
| 3 | Correctiveness | 6.2 | 6.0 | 5.2 |
| | Informativeness | 6.2 | 6.2 | 4.6 |
| 4 | Correctiveness | 8.8 | 8.0 | 4.8 |
| | Informativeness | 8.6 | 7.4 | 4.2 |
| 5 | Correctiveness | 7.6 | 7.4 | 6.4 |
| | Informativeness | 7.6 | 7.6 | 5.2 |

Table 5: Correctiveness and informativeness of the generated SFT data by human experts.

## A.2 CASE STUDY

In real-world urban planning scenarios, human planners spend much time dealing with textual data. We show two typical cases about urban planning text polishment and proofreading, to demonstrate the effectiveness of our tool in practical applications, as follows:

```
#prompt#
```

Assuming that you are a staff member who is good at writing and proofreading China's urban and rural planning texts, could you please polish and proofread the #original text# according to the #urban and rural planning text requirements#:

```
#urban and rural planning text requirements#
```

The text is coherent and complete. Standardized terminology, official and rigorous content, no colloquial expressions, no English, pinyin, internet terms, etc. In line with the style of government documents, the content is full, and there is no concept stacking and repetition of content. The content is reasonable and accurate, no misleading expressions, ensure consistency and logic. Coorect in grammar and spelling.

```
#original text#
```

It is difficult to find a historical and cultural district worthy of protection in a famous historical and cultural city, and the focus now is on restoring the traditional appearance of the historical city center as a whole.

```
#response#
```

A famous historical and cultural city currently has difficulty in finding a historical and cultural neighborhood worthy of protection. In order to enhance the historical and cultural value of the city, measures should be taken to enhance the protection of the existing cultural heritage, to identify and discover historical and cultural districts that have yet to be protected, and to restore the traditional appearance of the historical city center as a whole, so as to preserve and continue the historical and cultural characteristics of the city.

```
#prompt#
```

Assuming that you are a staff member who is good at writing and proofreading Chinese town and country planning texts, please touch up and proofread the #original text# according to the #urban and rural planning text requirements#.

```
#urban and rural planning text requirements#
```

The text is coherent and complete. Standardised terminology, official and rigorous content, no colloquial expressions, no English, pinyin, internet terms, etc. In line with the style of government documents, beautiful language, full of content, without concept stacking and repetitive content. Whether the content is reasonable and accurate, whether there are misleading expressions, to ensure consistency and logic. Pay attention to grammar, spelling and fluency.

```
#original text#
```
The layout of the city of Athens is a complete embodiment of the Hippodrome layout pattern, the Miletus is a layout pattern centred on the city square, with a grid of roads as the backbone , the square or the marketplace is surrounded by a series of public buildings, which are the core of urban life, the Acropolis has a very typical unplanned layout of features

```
#response#
```

The layout of the Athenian city does not fully reflect the Hippodrome layout pattern, while the Miletus is a typical layout pattern centred on the city square, with a grid of roads as the skeleton. A series of public buildings were constructed around the square or marketplace, which became the core of urban life. In addition, the Acropolis exhibits a typical unplanned layout characteristic, reflecting the diversity of history and culture.

```
(The above contents are translated from Chinese)
```

