# OpenReview forum: "UrbanPlanBench: A Comprehensive Assessment of Urban Planning Abilities in Large Language Models"
_ICLR.cc/2025/Conference — Submitted to ICLR 2025_

### Official Review · Reviewer_jSTs · 2024-10-30

**Soundness:** 2
**Presentation:** 3
**Contribution:** 2
**Rating:** 5
**Confidence:** 4

**Summary:**

The paper proposes UrbanPlanBench, a new benchmark with urban planning multiple choice questions to evaluate LLMs. Results on the benchmark show that most LLMs still fall short of urban planning. To improve LLMs’ performance on UrbanPlanBench, the authors experiment with RAG, CoT, as well as SFT using UrbanPlanText, an automatically curated corpus of urban planning knowledge.

**Strengths:**

1. The proposed UrbanPlanBench extends the popular LLM evaluation benchmarks with multiple-choice questions, such as MMLU and GPQA, to a new domain, urban planning. The benchmark is also in Chinese instead of English.

2. The authors conduct extensive experiments using different LLMs and attempt to enhance them with RAG, CoT, and SFT.

3. The paper is clear and easy-to-follow.

**Weaknesses:**

1. The quality of the proposed datasets needs to be discussed and further clarified.
- For UrbanPlanBench, the paper does not mention any engagement of experts from the corresponding discipline, i.e. urban planning, making this domain-specific benchmark less authoritative and credible.
- The annotation procedure of UrbanPlanBench is not described, such as how the urban planning qualification exams are accessed, how the exam questions are selected and adapted, if there are any cross-annotator validations to ensure the quality, and what is the annotator agreement of gold answers.
- Similarly, the authors did not include any information about how they design the expert evaluation for UrbanPlanText’s quality. It is unclear how the experts were instructed to give the “correctness” and “informativeness” scores, and how they relate to the quality of UrbanPlanText.
- If the question sources are available online, the authors should also discuss potential data contamination concerns and their impact on the evaluation of LLMs. For example, would the LLMs train on Chinese corpora have already seen the questions and answers so that they perform better?
- The authors claim to use UrbanPlanBench “to assess their [LLMs’] acquisition of planning skill” (line 71), but the benchmark is only testing LLMs’ domain knowledge. Suppose an LLM can answer most of the questions in the benchmark correctly, it is still not obvious how it can help in real-world urban planning tasks, such as predicting population growth and geographic analysis.

2. Several technical details are missing from the paper.
- The implementation of Self-RAG on different LLMs is not introduced. The statistics and examples of exam_1 and exam_2 corpora used for retrieval are not provided. The configuration, training process, and inference hyperparameters of Self-RAG are missing.
- The details of few-shot and zero-shot CoT prompting are missing, e.g. the number of in-context examples in few-shot, how they are annotated/retrieved, and prompt templates.
- While it is straightforward to use output probability to select the answers for MCQ-S, it is not explained how the authors apply this strategy to MCQ-M questions, where the questions have multiple answer choices in nondeterministic numbers.

**Questions:**

The paper mostly contains descriptive analysis of tables and figures, which focuses on the good results. It would be more inspiring for the community if the authors have any analysis explaining:

1. Why is MCQ-M much harder than MCQ-S, and the scaling of model hyperparmeters does not always lead to increased performance on MCQ-M?
2. Why does SFT not lead to consistent improvement across all LLMs and even decreased performance?
3. Have the authors conducted a systematic error analysis on one LLM to investigate the issues claimed in the paper, e.g. subject imbalance and language bias?

---

### Official Review · Reviewer_oDyp · 2024-11-01

**Soundness:** 2
**Presentation:** 2
**Contribution:** 2
**Rating:** 3
**Confidence:** 4

**Summary:**

The paper aims at advancing the LLM in the area of urban planning. The authors introduce a benchmark, UrbanPlanBench, that contains QAs on different perspectives of urban planning: (1) fundamental principles; (2) professional knowledge; (3) management and regulations. With the QAs, the authors evaluated the performance on various LLMs and settings (e.g., RAG, CoT) and found that current models still struggle in solving these tasks. In addition, the authors collect an SFT dataset named UrbanPlanText to help improve model performance.

**Strengths:**

1.	The topic is important and interesting. It would be great to see how LLMs can help improve the efficiency of experts in various domains.
2.	The experiments are comprehensive, with the evaluation of various models and settings.

**Weaknesses:**

1.	The size of the benchmark is too small. It is also limited to multiple-choice questions (no short open-domain questions). The reviewer is also unsure about the diversity and generalizability of the chosen questions. It seems that all the questions are from urban planning in China. However, according to S3 Management and regulations, such questions might not be applicable to urban planning in other countries. In Table 1, we observe higher performance of models from China, which strengthens my concern.
2.	The collection and quality control of the dataset is not well introduced. For example, what is the data source, human verification on the data correctness/categorization, and the inter-annotator agreement?
3.	The take-aways are blurred since there are not many experimental details introduced: e.g., how the model performance varies with hyperparameters such as temperature, how the RAG and CoT methods are designed, and most importantly, how the data is collected with quality control.
4.	From the reviewer’s point of view, there is a slight overclaim issue in the paper: this is not a benchmark on the general cross-culture urban planning with realistic scenarios that may assist real human experts. It is more like a QA dataset on the urban planning problem of some specific cultural background. The reviewer still acknowledges UrbanPlanText as a good contribution. It would be good if the authors could extend the benchmark to realistic settings that might help experts, beyond just question answering, e.g., retrieving useful cases.

**Questions:**

1.	Is it possible to show how the UrbanPlanText dataset helps other datasets (e.g., general question answering or reasoning)?
2.	What is the best way to evaluate the model performance in urban planning? What are the other potential tasks besides QA?
3.    How it the human annotation and experiments done? Can the authors provide more details on the annotation process, hyperparameter settings, actual prompts in CoT, retrieval settings, etc?

---

### Official Review · Reviewer_5qxn · 2024-11-02

**Soundness:** 2
**Presentation:** 3
**Contribution:** 2
**Rating:** 3
**Confidence:** 4

**Summary:**

The paper introduces UrbanPlanBench, a comprehensive benchmark for evaluating the efficacy of Large Language Models (LLMs) in urban planning. It also presents UrbanPlanText, the largest supervised fine-tuning (SFT) dataset for LLMs in this domain, comprising over 30,000 instruction pairs sourced from urban planning exams and textbooks. The benchmark and dataset aim to assess and enhance LLMs' capabilities in understanding urban planning principles, professional knowledge, and management regulations. The paper reveals significant room for improvement in LLMs' performance, particularly in tasks requiring domain-specific terminology and reasoning.

**Strengths:**

1. This work proposes a benchmark that covers multiple dimensions of urban planning and accordingly constructs a dataset for fine-tuning and enhancing model performance.
2. This work demonstrates that most models do not achieve human-level performance on urban planning tasks, while also employing methods to enhance the models' capabilities in urban planning.

**Weaknesses:**

1. UrbanPlanBench seems to focus solely on processing data from the 2022 urban planning qualification exam, which limits its contributions:
    - The authors converted the original 2022 qualification exam documents into csv format, creating the UrbanPlanBench, which comprises 300 MCQs. Could you add a brief discussion highlighting your contributions to the benchmark beyond the data processing steps?
    - Additionally, since the evaluation data is sourced from publicly available texts, it is difficult to ensure that large models did not encounter this data during pre-training. This could undermine the usability of UrbanPlanBench and affect fair comparisons between models. Moreover, as the data is in Chinese, models like Qwen, trained on larger Chinese corpora, show better urban planning performance, raising concerns about whether they have already been trained on these data.
2. The experiments may be insufficient:
    - In Sec 2.3, this paper investigates prompting techniques, including RAG and CoT. In Sec 3, this paper evaluates how fine-tuning methods could enhance the capabilities of LLMs. However, while the former is based on GPT-4o-mini and the latter on other LLMs, their results are not comparable. What we may really care about is the comparison between these two types of methods.
    - The paper mentions "a challenge in sourcing relevant data for SFT". Given the difficulty in obtaining this data, why use SFT to enhance the models' capabilities if prompting techniques, such as RAG and CoT, have been proved effective?
3. The effectiveness of the SFT methods:
    - After SFT on UrbanPlanText, as illustrated in Table 4, 70%, 50% and 40% of LLMs exhibited decreased accuracy on the full questions of S1, S2 and S3, respectively. However, the models after SFT do not show performance improvements on many test sets.
4. Other weaknesses:
    - There is a typo in Line 284 and 464: "MCS-S" should be "MCQ-S".

**Questions:**

1. Why was multiple-choice questions (MCQ) chosen as the primary evaluation format when designing UrbanPlanBench? Have other types of assessment methods been considered?
2. How does the author plan to address potential biases in the benchmark and dataset, particularly considering that all the data comes from Chinese urban planning exams and that some of the evaluation data may have been learned during the model's pre-training phase?
3. Why is the SFT effect minimal on larger models? Does this indicate that the quality of the constructed training data is poor?

---

### Official Review · Reviewer_47Ea · 2024-11-09

**Soundness:** 2
**Presentation:** 2
**Contribution:** 2
**Rating:** 3
**Confidence:** 4

**Summary:**

This paper introduces PlanBench, a benchmark designed to assess the effectiveness of LLMs in the field of urban planning. The study finds notable gaps in LLMs' planning knowledge, with 70% of models performing poorly in regulatory understanding. Additionally, the authors present PlanText, the largest supervised fine-tuning dataset for LLMs in urban planning, containing over 30,000 instruction pairs from exams and textbooks. Fine-tuned models show improved performance but still struggle with domain-specific terminology and reasoning tasks. The benchmark, dataset, and tools aim to drive LLM integration into urban planning, enhancing collaboration between human expertise and AI.

**Strengths:**

The paper presents a valuable dataset focused on urban planning, an essential domain for AI exploration. The structure is thorough, covering dataset construction, in-depth analysis, and the training method. Additionally, the writing is clear and easy to follow, making the paper accessible and well-organized.

**Weaknesses:**

While this benchmark introduces a new domain, urban planning, it doesn't fundamentally expand beyond the scope of existing benchmarks. Its formulation is quite similar to widely used evaluation benchmarks like BigBench and MMLU, with the primary difference being just the new domain focus. This makes the dataset's contribution feel less novel, as it isn't that different from prior works. Similarly, the training method lacks novelty, as it relies on leveraging domain-specific resources from the web, which is already a common practice.

To enhance the paper, the benchmark's evaluation could move beyond a multiple-choice QA format. A more realistic approach might involve simulating a full urban planning task: for example, an agent would plan steps for constructing a house or shopping mall in a given city or district, considering the environment state in a simulated setting. This could yield more interesting insights into agent performance, reveal complex failure patterns, and potentially offer valuable contributions to real-world urban planning.

**Questions:**

See above

---

### Meta-Review · Area_Chair_VXKT · 2024-12-22

**Metareview:**

The paper introduces UrbanPlanBench, a benchmark for evaluating LLMs in urban planning, along with UrbanPlanText, a dataset of 30,000 instruction pairs sourced from urban planning exams and textbooks. While fine-tuning shows some improvement, LLMs continue to struggle with tasks requiring specialized reasoning and terminology. UrbanPlanBench opens up a new domain for LLM evaluation, and UrbanPlanText provides a useful fine-tuning resource. However, reviewers pointed out that the benchmark lacks novelty, adapting existing frameworks, and relies on a small dataset focused on Chinese urban planning exams, which limits diversity and generalizability. Reviewers concerned about potential data contamination during pre-training and weak annotation practices, which hurt the dataset’s credibility. The reliance on multiple-choice questions might restrict real-world applicability, and fine-tuning results were inconsistent, especially for larger models. While the paper takes an interesting step forward, expanding beyond multiple-choice questions, and involving domain experts for validation would make the benchmark more strong. I would recommend to further refine and recycle the paper.

**Additional Comments On Reviewer Discussion:**

Reviewers raised concerns about the limited novelty of the benchmark, dataset quality, and potential data contamination.  Concerns about the benchmark’s narrow focus on multiple-choice questions and the inconsistent results of fine-tuning remained unresolved. Reviewers also questioned the generalizability of the benchmark, given its focus on Chinese urban planning, and suggested expanding the scope and incorporating more realistic tasks. The author didn't respond to the review.

---

### Decision · Program_Chairs · 2025-01-22

Reject